# iiANET: Inception Inspired Attention Hybrid Network for efficient Long-Range Dependency

**Yunusa Haruna**                                                   *yunusa2k2@buaa.edu.cn*
*NewraLab, Suzhou, China*

**Adamu Lawan**                                                      *alawan@buaa.edu.cn*
*Beihang University, Beijing, China*
*NewraLab, Suzhou, China*
*Beijing GoerTek Alpha Labs*

**Abdulganiyu Abdu Yusuf**                                          *abdulg720@gmail.com*
*Beijing Institute of Technology, Beijing, China*

**Reviewed on OpenReview:** *https://openreview.net/forum?id=HGSjlgFodQ*

## Abstract

The recent emergence of hybrid models has introduced a transformative approach to computer vision, gradually moving beyond conventional convolutional neural networks and vision transformers. However, efficiently combining these two approaches to better capture long-range dependencies in complex images remains a challenge. In this paper, we present iiANET (Inception Inspired Attention Network), an efficient hybrid visual backbone designed to improve the modeling of long-range dependencies in complex visual recognition tasks. The core innovation of iiANET is the iiABlock, a unified building block that integrates a modified global r-MHSA (Multi-Head Self-Attention) and convolutional layers in parallel. This design enables iiABlock to simultaneously capture global context and local details, making it effective for extracting rich and diverse features. By efficiently fusing these complementary representations, iiABlock allows iiANET to achieve strong feature interaction while maintaining computational efficiency. Extensive qualitative and quantitative evaluations on some SOTA benchmarks demonstrate improved performance.

## 1 Introduction

From autonomous drones to urban planning, understanding complex visual scenes is more critical than ever, yet traditional models struggle to capture such complexity. Over the last decade, deep Convolutional Neural Network (CNN) architectures have emerged as the de facto standard for solving most computer vision (CV) tasks, including image classification He et al. (2016); Tan & Le (2019), object detection Ren et al. (2015); Redmon & Farhadi (2017) and segmentation Long et al. (2015) with compelling results. The prevalence of CNN architectures is not coincidental, as they excel at capturing spatial features and patterns in images. However, the dominance of CNN architectures is being challenged by the emergence of ViT (Vision in Transformer) Dosovitskiy et al. (2020), presenting a transformative approach to solving CV tasks. Interestingly, this groundbreaking model outperforms SOTA CNN-based models on ImageNet benchmark Dosovitskiy et al. (2020) and emerges as a competitive alternative Han et al. (2022). Practically, ViT works exactly like the text-based Natural Language Processing (NLP) transformers but with patch embedding. It divides the input image into patches, projects them into a high-dimensional feature space through a linear projection layer, adds positional embedding, passes them through a transformer encoder, and finally maps the output to a fixed-length vector for classification tasks.

Significantly, the key component of ViT is the self-attention mechanism Dosovitskiy et al. (2020) within the encoder, which enables the model to capture long-range dependencies by allowing each element in the input sequence to attend to all other elements, considering their relative importance Dosovitskiy et al. (2020). While this capability allows the model to selectively focus on distantly related pixels, facilitating the efficient capture of contextual information across the entire input sequence, it encounters limitations such as increased computational complexity, reduced interpretability, data hungry, and challenges in handling spatial information effectively compared to CNNs. In contrast, CNN-based models, while effective at capturing local features through parameter sharing and local receptive fields, struggle with capturing long-range dependencies, limiting their ability to integrate distant pixel relationships. These limitations have led to the development of hybrid models, which combine their strengths to improve performance Haruna et al. (2025).

Specifically, previous hybrid designs have aimed to enhance capturing long-range dependencies for various CV tasks Guo et al. (2022); Srinivas et al. (2021); Dai et al. (2021a). However, the design of hybrid models introduces additional design complexities Haruna et al. (2025), and computational costs compared to monolithic models Khan et al. (2023), while also potentially leading to information loss due to feature fusion of distinct models Haruna et al. (2025). Lastly, more effort is needed to design efficient hybrid models capable of capturing long-range dependencies in complex images, a challenge that remains largely unaddressed.

In this work, we propose a novel architectural block, the iiABlock (Inception-Inspired Attention Block), which serves as the core component of our model, iiANET. The iiABlock is a carefully designed hybrid module that integrates parallel convolutional layers for efficient local feature extraction, and a global 2D Multi-Head Self-Attention (MHSA) mechanism with Registers to effectively model long-range dependencies. The outputs from these branches are then fused via concatenation and feature shuffling, enabling rich interaction between local and global features. By leveraging the complementary strengths of CNNs and transformers in a lightweight design, iiANET offers a simple yet powerful solution for understanding complex visual scenes with long-range dependencies. For example, on the AID (Aerial Image Dataset) Xia et al. (2017), iiANET-B and iiANET-L achieve an accuracy of 80.57% and 83.11% respectively, outperforming ResNet-50 (71.93%), ViT-B/224 (69.93%), and DiNAT-B (79.12%). These results highlight iiANET's effectiveness in modeling long-range dependencies in challenging datasets. The contributions of this paper are summarized as follows:

- From a methodological perspective, we identify that existing vision models struggle to efficiently capture long-range dependencies and global context in complex scenes, creating a gap in robust visual understanding.

- We introduce iiABlock, a novel hybrid module that integrates parallel convolutional branches with global rMHSA, enabling efficient capture of long-range dependencies in complex vision tasks.

- Extensive experimental results on commonly used benchmarks demonstrate that iiANET outperforms some existing SOTA methods.

## 2 Related Work

CNN-based methods have seen various attempts to enhance their ability to capture LRD in images. Donahue et al. (2015) introduced the Long-term Recurrent Convolutional Network (LRCN) by fusing CNNs with LSTMs, while Yu et al. (2017) proposed the Dilated Residual Network (DRN) using multiple dilation rates to expand receptive fields, and Yu & Koltun (2015) designed a Dilated Convolution (DC) model to improve global context in semantic segmentation. ADRnet augments CNN with advection, diffusion, and reaction terms to enable non-local feature transport and improve LRD spatio-temporal modeling Zakariaei et al. (2024). These approaches advanced CNN capacity for LRD but face limitations: LRCN increases computational complexity due to recurrent connections, DRN can lose fine-grained spatial details from varying dilation rates, and DC suffers from gridding artifacts. The emergence of ViT Dosovitskiy et al. (2020); Liu et al. (2021a) offered a breakthrough in capturing LRD via attention mechanisms, achieving SOTA performance. However, their quadratic complexity, high data requirements, and weaker inductive bias compared to CNN demand substantial computational resources. To address these trade-offs, hybrid methods combine CNN feature extraction with ViT global dependency modeling Haruna et al. (2025). Zhang et al. (2022) proposed ELAN, using group-wise multi-scale self-attention for super-resolution; Guo et al. (2022) introduced

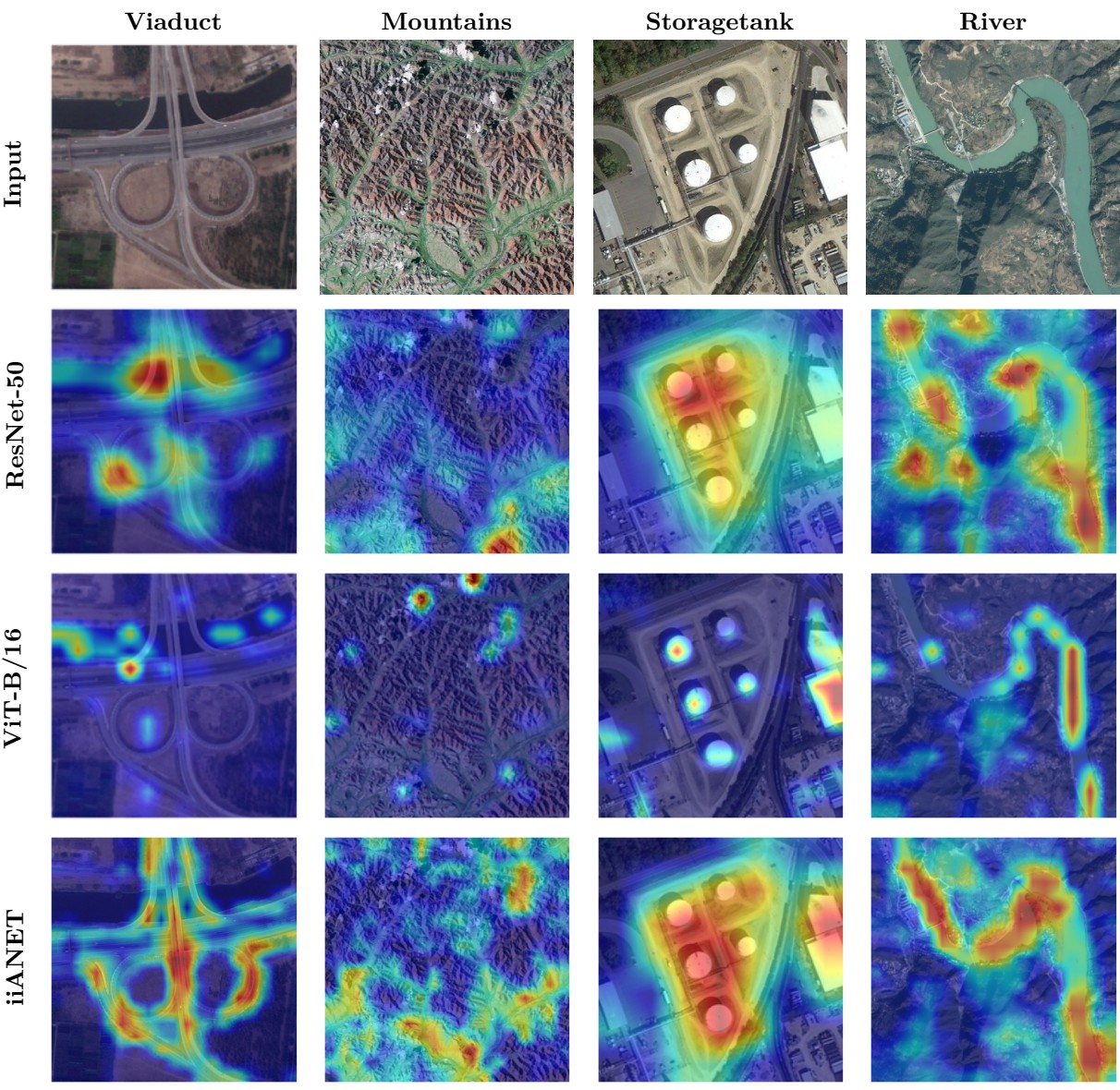

Figure 1: iiANET Grad-CAM Selvaraju et al. (2017) comparison and other state-of-the-art models, e.g., (a) shows an aerial image of viaduct, mountain, storage tanks, and river featuring complex infrastructure consisting of multiple spans, roads, and surrounding landscapes. The primary objective is to accurately detect and classify various elements to facilitate efficient maintenance, safety management, and infrastructure planning. Consequently, capturing long-range dependencies in this scenario is crucial for comprehending the spatial layout of different viaducts, mountains, storage tanks, and rivers, their interactions, and potential structural issues. (b) ResNet-50 highlights strong local features but fails to capture long-range dependencies. (c) ViT-B/16 demonstrates limited interpretability, primarily focusing on small, scattered regions. (d) The proposed iiANET (hybrid model) exhibits enhanced ability to capture long-range dependencies, global context, and improved interpretability.

CMT (CNN Meet ViT) to integrate attention into CNN blocks; and Srinivas et al. (2021) developed BoTNet by replacing the final ResNet block with MHSA. While effective, these methods often apply attention at later stages with smaller spatial dimensions, limiting effectiveness, and face challenges such as memory constraints (CMT-L), structural complexity, and information loss from fusing distinct methods Peng et al. (2021); Dai et al. (2021b); Hassani & Shi (2022); Wu et al. (2021). CNN excel at local detail capture but struggle with LRD Haruna et al. (2025); Khan et al. (2023), RNNs handle such dependencies but lack parallelism and train slowly Banerjee et al. (2019), and ViT capture them efficiently but require more memory Dosovitskiy et al. (2020). However, it remains a challenge to efficiently combine CNN and ViT architectures due to design complexity, higher computational costs, feature fusion losses, and interpretability issues. This paper addresses this gap by proposing a hybrid model that efficiently captures LRD in complex images.

## 3 Method

### 3.1 Our Approach: iiABlock

The iiABlock is the core component of the proposed iiANET, designed to capture both local and global features in complex visual scenes. It combines parallel convolutional layers for efficient extraction of localized features with a global 2D r-MHSA mechanism augmented by Register tokens to model long-range dependencies. The convolutional branch leverages parameter sharing and local receptive fields, while 2D r-MHSA attends to distant pixel relationships across the input, with Register tokens to enhance interpretability. To merge these complementary features, we introduce a lightweight fusion strategy using feature concatenation followed by channel shuffling, enabling rich local–global interactions with minimal computational cost. This balanced design offers a favorable trade-off between speed and accuracy, providing a robust backbone for downstream visual recognition tasks. Figure 2 illustrates the iiABlock architecture.

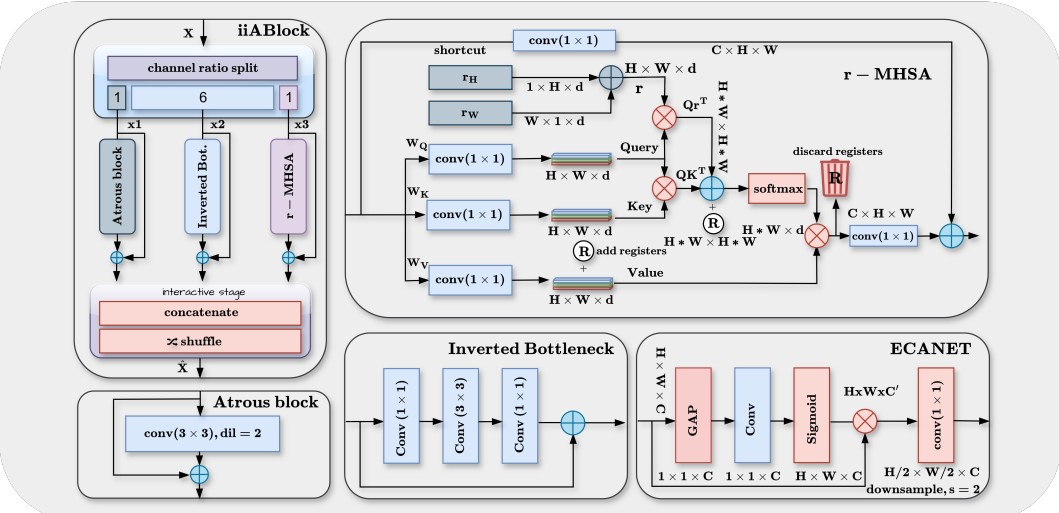

Figure 2: iiABLOCK design showing r-MHSA, inverted bottleneck, ECANET and Atrous block

### 3.2 Local Details

To extract fine-grained features and spatial patterns from complex images, iiABlock introduces components for modeling local details. This is important for recognizing textures, edges, and region-specific patterns.

**Inverted Bottleneck** *(Efficient Convolutional Block)* iiABlock utilizes the inverted bottleneck in parallel to improve computational efficiency and enhance local feature extraction, consisting of 1×1 convolutions for dimensionality reduction, 3×3 depth-wise separable convolution for spatial information extraction, and 1×1 convolution for projection. Notably, this block is limited to capturing local context with a fixed kernel,

making it less effective in understanding prevalent global context in complex images Dai et al. (2021a), e.g., road, viaduct, bridge. Given the depth-wise operation in equation 1.

$$y_i = \sum_{j \in \mathcal{L}(i)} w_{i-j} \cdot x_j \tag{1}$$

$y_i$ calculates the output at position $i$ by taking a weighted sum of the input $x_i$, where $x_i, y_i \in \mathbb{R}^D$. The weights $w_{i-j}$ determine the contribution of each input $x_j$ to the output, and $\mathcal{L}(i)$ represents a local neighborhood, typically a $3 \times 3$ grid centered around $i$. The small size of $\mathcal{L}(i)$ limits the receptive field's ability to capture intricate details, particularly in complex images with prevalent long-range dependencies. To mitigate the limitation of the inverted bottleneck in capturing long-range dependencies, we also introduce atrous convolution into the iiABlock.

**Atrous Convolution** *(Expanding Receptive Field)* In iiABlock, a single $3 \times 3$ atrous convolution expands the receptive field without increasing parameters. Unlike standard convolution, it applies a dilation rate $r$ to space kernel elements, covering a wider area while preserving resolution and computational efficiency (See equation 2. Given an input feature map $x$ and filter $w$, the output at location $i$ is:

$$y[i] = \sum_{k=0}^{K-1} x[i + r \cdot k] \cdot w[k] \tag{2}$$

This captures mid-range dependencies and enriches contextual understanding, bridging local and global representations. However, it remains insufficient for fully modeling global, long-range dependencies in complex scenes, which are further addressed by integrating a global r-MHSA module into iiABlock.

### 3.3 Global Details

**r-MHSA** *(Capturing global context and long-range dependencies)*. To capture global context and long-range dependencies in complex images, iiABlock integrates a modified global 2D r-MHSA mechanism. Unlike CNN layers limited to local receptive fields, r-MHSA allows each spatial location to attend to all others, effectively modeling contextual relationships across the entire image. Given a 2D input feature $X \in \mathbb{R}^{C \times H \times W}$ reshaped into $X \in \mathbb{R}^{HW \times d}$ (where $d$ is the feature dimension), linear projections generate queries $Q = XW_Q$, keys $K = XW_K$, and values $V = XW_V$. Attention $Z$ with $h$ heads is computed as shown in Equation 3, enabling each token to attend to all others and capture long-range dependencies.

$$Z_h(Q_i, K, V) = \text{softmax}\left(\frac{Q_i K^\top}{\sqrt{d_k^h}}\right) V \tag{3}$$

Here, $Q_i$ interacts with all keys in $K$ across the entire sequence, unlike the standard MHSA, which attends within a limited context window. This enables the model to consider the relationships between all tokens regardless of their positional distance, capturing the global context and long-range dependencies prevalent in complex images. The `softmax` operation normalizes these scores and produces attention weights, which, when applied to the value matrix $V$, compute the final attended values. However, this interaction is order-agnostic and doesn't capture positional relationships in the input sequence. Therefore, in image data where spatial information is essential, integrating positional encodings is necessary to effectively complement MHSA.

**Relative Position Encoding Srinivas et al. (2021)** MHSA is permutation equivariant with no positional encoding. This characteristic limits its representational power, particularly for vision tasks involving highly structured data like images. Notably, it is added to the input image representation before the MHSA is applied, and it is used to guide the attention weights to focus on relevant pixels based on their relative positions in the input image.

$$Z_h(Q_i, K, V) = \text{softmax}\left(\frac{Q_i K^\top + Q_i r}{\sqrt{d_k^h}}\right) V \tag{4}$$

Where $r$ is a trainable matrix. Lastly, reshape $Z(X)_h$ back to its original spatial shape of $X \in \mathbb{R}^{C \times H \times W}$. This addresses MHSA's order-agnostic nature, enhancing its representational power.

**Registers** *(Improving interpretability)*. While the self-attention mechanism significantly improves the network's ability to capture long-range dependencies, it struggles with poor interpretability Darcet et al. (2023). We instead add additional learnable tokens to mitigate prevalent artifacts in the attention mechanism caused by high norms in image areas with low information during inference or training, similar to the implementation by Darcet et al. (2023). In this case, it is a global MHSA where the attention mechanism has a single input image, in contrast to having several patches. We initialize the register tokens for queries and keys as $R_{QK} \in \mathbb{R}^{N \times HW \times HW}$, where $N$ is the number of register tokens and $HW$ is the spatial dimension, then value register tokens as $R_V \in \mathbb{R}^{N \times \frac{D}{\text{head}} \times HW}$, where $D$ is the dimension. The operations expand both $R_{QK}$ and $R_V$ to $\mathbb{R}^{B \times N \times H \times W}$ and $\mathbb{R}^{N \times \frac{D}{\text{head}} \times HW}$, respectively (equation 5 and 6) effectively creating $B$ copies of the register tokens for each batch, where $B$ is the batch size. This step ensures that each batch has its own set of register tokens, facilitating batch-wise parallel processing in the attention mechanism.

$$R_{QK} = \text{repeat}\big(R'_{QK}, \ nhw \to bnhw', \ b = B\big) \tag{5}$$

$$R_V = \text{repeat}\big(R'_V, \ nhw \to bnhw', \ b = B\big) \tag{6}$$

Then integrate the register tokens into the attention mechanism as $Q_i K_R^\top = Q_i K^\top + R_{QK}$ and $V_R = V + R_V$ before computing the attention, where $Z_R$ is the final 2D-MHSA output with registers. After computation, the register tokens are discarded (See equation 7).

$$Z_R^h(Q_i, K, V) = \text{softmax}\left(\frac{Q_i K_R^\top + Q_i R}{\sqrt{d_k^h}}\right) V_R \tag{7}$$

### 3.4 Features Recalibration

**ECANET** *(Channel-wise Recalibration and Down-Sampling)*. While long-range dependencies in computer vision span both spatial and channel dimensions, traditional CNNs and attention mechanisms often emphasize spatial adaptivity, neglecting channel-wise adaptability. To address this, iiABlock integrates ECANET Wang et al. (2020), which efficiently computes channel attention weights by modeling inter-channel relationships. Compared to SENET Hu et al. (2018), ECANET offers improved efficiency, scalability, and accuracy. Given input $X \in \mathbb{R}^{C \times H \times W}$, adaptive average pooling reduces spatial dimensions to $P \in \mathbb{R}^{C \times 1 \times 1}$, followed by a $1 \times 1$ convolution and sigmoid activation to produce attention weights $Z \in [0, 1]$. These weights modulate channel significance through element-wise multiplication: $ECA = X \otimes Z$. Feature maps are then downsampled via a $1 \times 1$ convolution with stride 2 after each stage.

### 3.5 Feature Interaction Fusion

iiABlock enables efficient multi-scale feature learning through a structured feature interaction mechanism that splits input channels into three branches: atrous convolution, inverted bottleneck, and r-MHSA. Each branch processes its feature subset independently and the outputs are fused via concatenation, followed by channel shuffling to enhance cross-branch interaction. This design captures short-range, long-range, and channel-specific dependencies in parallel while maintaining simplicity and low computational cost, outperforming complex fusion strategies such as heavy cross-attention Chen et al. (2021a) or multi-level stacking Li et al. (2019). Empirically, we found the channel ratio $r = (1\!:\!6\!:\!1)$ optimal. For input $X \in \mathbb{R}^{C \times H \times W}$, channels are split as $x_1, x_2, x_3$ with dimensions approximately $\lfloor C/8 \rfloor$, $\lfloor 3C/4 \rfloor$, and $\lfloor C/8 \rfloor$, respectively. Corresponding functions $f_1$, $f_2$, and $f_3$ process each branch, whose outputs are concatenated (Figure 3). Finally, channel shuffling is applied to strengthen inter-branch interaction, defined in equation 8:

$$\hat{X} = \text{shuffle}(f(x)) \in \mathbb{R}^{C \times H \times W}. \tag{8}$$

The fused output $\hat{X}$ effectively combines diverse receptive fields and attention mechanisms with efficiency and expressiveness. Table 1 presents the schematic details of iiABlock.

Table 1: Schematic of spatial sizes and channel allocations across branches and fusion.

| Branch | Input Size | Operation | Output Size |
|---|---|---|---|
| r-MHSA | $\frac{C}{8} \times H \times W$ | r-MHSA | $\frac{C}{8} \times H \times W$ |
| Inverted Bottleneck | $\frac{3C}{4} \times H \times W$ | Convolution | $\frac{3C}{4} \times H \times W$ |
| Atrous Conv | $\frac{C}{8} \times H \times W$ | Atrous Conv | $\frac{C}{8} \times H \times W$ |
| Fusion | $(C/8 + 3C/4 + C/8) \times H \times W$ | $\text{concat}(y_1, y_2, y_3) \to \text{shuffle}$ | $C \times H \times W$ |

## 3.6 Computational-Efficiency

iiABlock processes visual inputs $X \in \mathbb{R}^{C \times W \times H}$ (with $M = W \cdot H$) using r-MHSA, dilated, and inverted bottleneck convolutions. Given an expansion factor $E = 2C$, their computational complexities are:

$$\Omega(\text{MHSA}) = 4MC^2 + 2M^2C, \tag{9}$$

$$\Omega(\text{DilatedConv}) = 18MC^2, \tag{10}$$

$$\Omega(\text{Bottleneck}) = MC^2 \left( \frac{2}{r} + \frac{9}{r^2} \right), \tag{11}$$

where $r$ is the channel reduction ratio. To enhance global modeling, iiANET adds Register Tokens to the r-MHSA branch. For $N$ register tokens per head, the cost becomes:

$$\Omega(\text{MHSA+Reg}) = 4(M + N)C^2 + 2(M + N)^2C, \tag{12}$$

where $N \ll M$, making the added cost negligible while improving stability and representation quality. Overall, r-MHSA captures global context but scales quadratically with $M$, whereas dilated and bottleneck convolutions scale linearly, offering efficient local feature extraction. iiABlock balances both for scalable high-resolution performance.

## 3.7 Memory-Efficiency

To reduce memory consumption and computational cost, we allocate the r-MHSA branch only 1/8 of the input channels. Additionally, r-MHSA is placed deep in the network where the spatial dimensions are low, mitigating the quadratic scaling with sequence length $M$. This design ensures that global attention is captured effectively without excessive memory usage, allowing iiABlock to process high-resolution images efficiently.

## 3.8 iiANET Architectural Overview

iiANET is a hybrid visual recognition backbone architecture designed to enhance capturing long-range dependencies in complex images while maintaining computational efficiency. At each stage, iiANET stacks iiABlock in parallel across four stages. Each iiABlock captures both short-range and long-range dependencies while incorporating global context, enabling the model to process intricate patterns effectively. The architecture is also non-isotropic as it down-samples spatial features at every stage, allowing for a progressive abstraction of features. By stacking iiABlock in parallel at each stage, iiANET efficiently captures both fine-grained local details and global context, making it well-suited for complex vision tasks. Figure 3 illustrates iiANET's architecture.

**Stem** *(Initial stage)* Given the higher resolutions of complex images, this component serves to compress the computational costs of iiANET by shrinking the spatial dimensions of the input image to half, trading off spatial details for improved model efficiency and basic feature extraction Bello et al. (2021). Let the input image be $X \in \mathbb{R}^{3 \times H \times W}$, we apply two sequential $3 \times 3$ convolutional layers, each followed by a batch normalization and ReLU activation function, with the initial layer using a stride of 2.

**iiABlock** *(Building Blocks)* The iiANET-B and iiANET-L variants stack iiABlocks in non-isotropic configurations of $[2, 3, 5, 2]$ and $[2, 3, 10, 4]$ across four stages to capture multi-scale features while reducing spatial

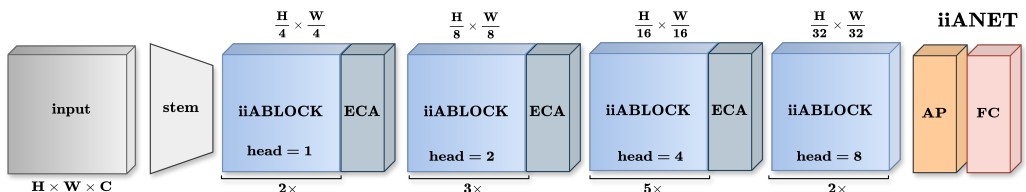

Figure 3: iiANET architectural overview.

resolution. At each stage, input feature maps are down-sampled, yielding spatial dimensions of $H/4 \times W/4$, $H/8 \times W/8$, $H/16 \times W/16$, and $H/32 \times W/32$. Inspired by Inception designs Szegedy et al. (2015), multiple iiABlocks run in parallel per stage, enabling simultaneous processing of features with varied receptive fields. These branches are fused via concatenation followed by channel shuffling to promote cross-path interaction. This hierarchical, multi-branch structure allows iiANET to efficiently capture short and long-range dependencies while maintaining a compact, efficient architecture suited for complex vision tasks.

**Output Layer**. After the final stage, Adaptive Pooling reduces the feature map to a fixed size, which is then passed through a Fully Connected layer for classification.

## 4 Experimental Results and Comparisons

We evaluate iiANET qualitatively and quantitatively on several widely used benchmark datasets, comparing it with some state-of-the-art CNN, ViT, SSM and hybrid models. The evaluation covers classification performance and effectiveness as a backbone for object detection and segmentation, focusing on iiANET's ability to capture long-range dependencies in complex images.

**Datasets and Metrics.** Experiments are conducted on diverse datasets: AID Xia et al. (2017) (10,000 images, 30 scene classes), Oxford-IIIT Parkhi et al. (2012) (4,978 images, 37 cat and dog breeds), and RLD Sethy et al. (2020) (5,932 images, 4 disease categories). Additionally, ImageNet1K Deng et al. (2009) (1.28M images, 1,000 classes), COCO-2017 Lin et al. (2014) (118K training images, 80 object categories), and ADE20K Zhou et al. (2017) (25K images, 150 semantic categories) are employed to assess generalization and robustness. Metrics include top-1/top-5 accuracy, Average Precision (AP), FLOPs, and throughput, providing comprehensive performance insights.

**Experimental Setup**. Training was performed on a Linux system with an Intel Core i7-8700K CPU, 2 NVIDIA Titan XP GPUs (12GB), and 32GB RAM. Models were trained for 90 or 150 epochs with batch size 16 using the AdamW optimizer, an initial learning rate of 0.0001, and a decay rate of 0.05. For fair comparison, some baseline models were re-trained on AID, Oxford-III, and RLD using the authors' default settings, while models with reported results were taken from the original publications to ensure fairness and consistency, as full re-training under a unified configuration would be computationally infeasible.

### 4.1 Qualitative Evaluation and Comparison: iiANET Visual Inspection

We applied Grad-CAM Selvaraju et al. (2017) on the final layer of iiANET and several state-of-the-art models ResNet-50/100 He et al. (2016), EfficientNet-B4/B5 Tan & Le (2019), DenseNet-169/201 Huang et al. (2017), ViT-B/L-16 Dosovitskiy et al. (2020), CoatNet-3 Dai et al. (2021a), and BoTNet Srinivas et al. (2021) with visualizations shown in Figure 4. CNN-based models generate localized heatmaps around objects due to their limited receptive fields, while ViT-based models produce scattered attention spots, indicating interpretability challenges. Hybrid models better capture long-range dependencies and improve interpretability. Notably, iiANET excels at precisely outlining complex objects with minimal background noise, suggesting enhanced accuracy and reliability in tasks reliant on long-range dependencies, such as medical imaging, autonomous driving, remote sensing, and security surveillance.

### 4.2 Qualitative Results on Detection and Segmentation

Figure 5 presents qualitative results on object detection, instance segmentation, and semantic segmentation tasks. Specifically, we evaluate iiANET as a backbone within Faster R-CNN  Ren et al. (2015) and Mask R-CNN  He et al. (2017) on the COCO val2017 dataset Lin et al. (2014), and UperNet  Xiao et al. (2018) for semantic segmentation on ADE20K Zhou et al. (2017).  The visual results demonstrate that iiANET effectively generalizes across different dense prediction tasks, capturing both fine-grained object boundaries and global contextual cues.  This shows that iiANET can be seamlessly integrated into standard detection and segmentation frameworks while maintaining quality visual predictions.

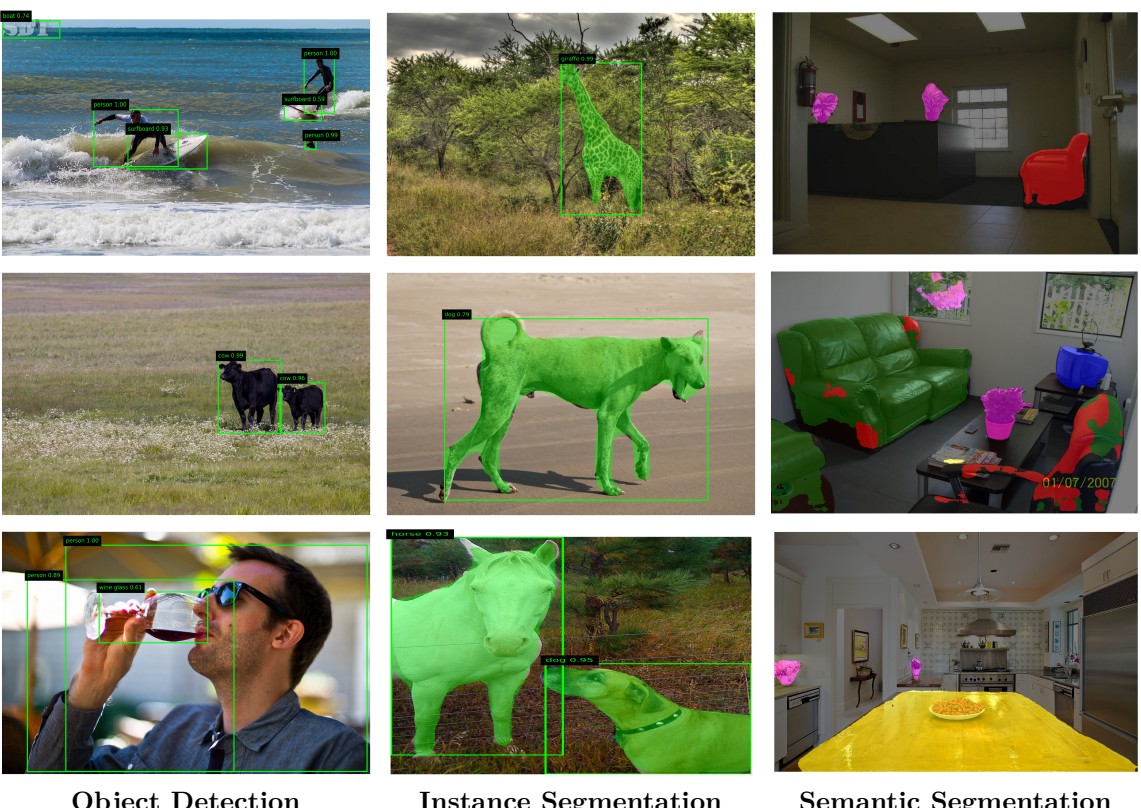

**Object Detection**     **Instance Segmentation**     **Semantic Segmentation**

Figure 5: Qualitative results of object detection and instance segmentation on COCO val2017 Lin et al. (2014), and semantic segmentation on ADE20K Zhou et al. (2017). From left to right, the results are obtained using iiANET as the backbone in Faster R-CNN, Mask R-CNN, and UperNet for semantic segmentation.

### 4.3 Quantitative Evaluation and Comparison

### 4.3.1 Classification performance

Table  2 shows iiANET classification performance across three standard benchmark datasets. The quantitative evaluation involves comparing iiANET-B and iiANET-L with several recent state-of-the-art models in terms of both accuracy and computational cost on ImageNet-1K, AID and Oxford-III.

- **ImageNet-1K:** iiANET-L achieves a competitive top-1 accuracy of 84.9% while maintaining computational efficiency with only 13.45 GFLOPs and 50.9M parameters.  This demonstrates a decent accuracy-efficiency tradeoff, outperforming larger and more computationally expensive models such as ViT-L/16 (76.5%, 59.69 GFLOPs) and CoAtNet-3 (84.5%, 32.53 GFLOPs).  These results show iiANET's ability to learn robust, discriminative features with minimal complexity.

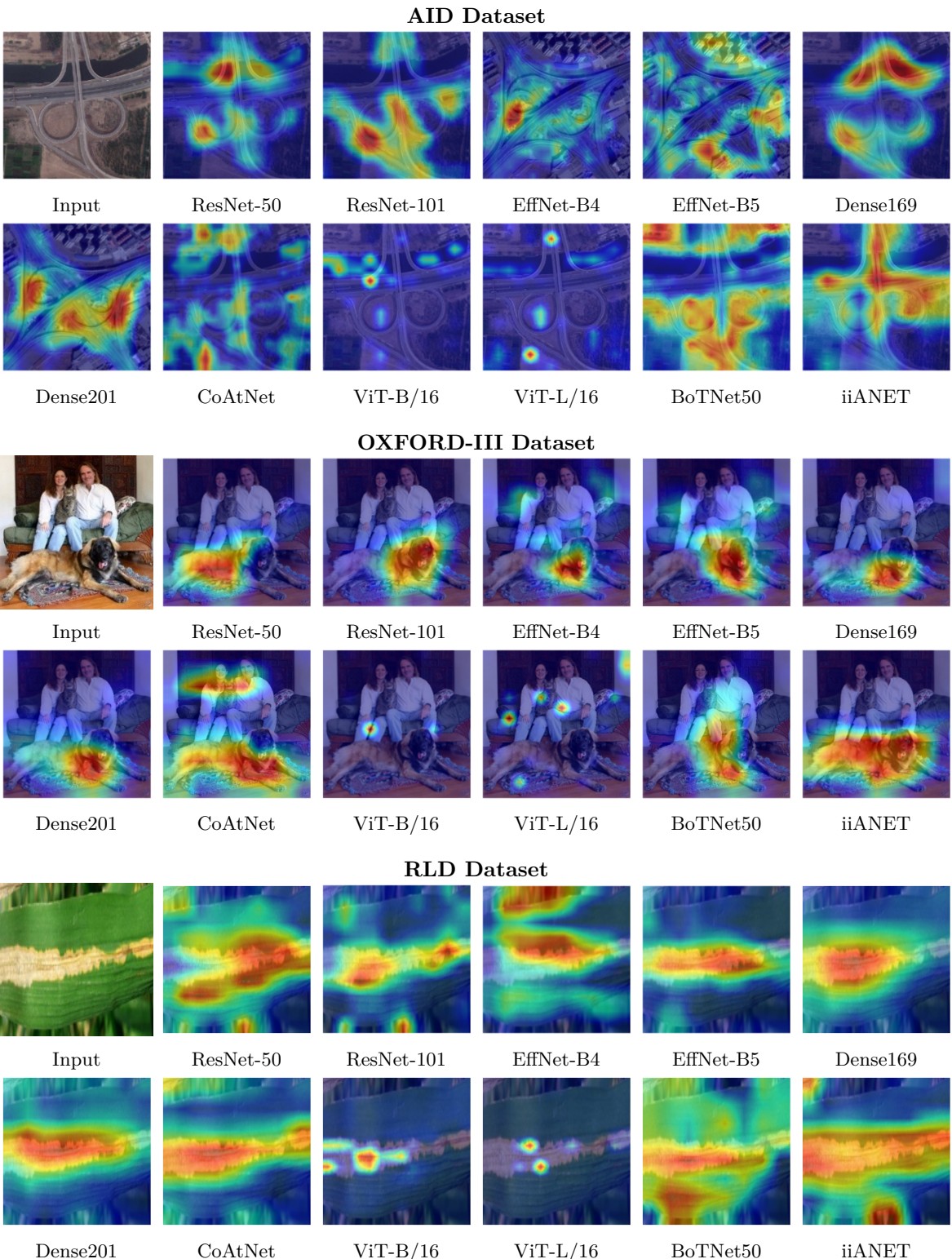

Figure 4: Visual inspection of iiANET compared to some SOTA models using Grad-CAM Selvaraju et al. (2017) highlights the model's ability to focus on complex object regions. The heatmaps demonstrate iiANET's improved capacity to capture long-range dependencies and provide more interpretable attention on relevant spatial structures compared to ResNet-50, BoTNet, ViT-B/16, and CoAtNet.

- **AID Dataset:** Contains complex spatial structures and long-range dependencies, which challenge traditional models. iiANET-B shows good performance, achieving 80.57% top-1 accuracy using only 8.22 GFLOPs. This surpasses some established CNN-based backbones such as ResNet-101 (68.93%) and DenseNet-201 (71.60%), as well as some transformer-based models like ViT-B/16 (69.93%), indicating iiANET's enhanced ability for modeling spatial complexity in aerial imagery.

- **Oxford-IIIT Pets:** In the fine-grained Oxford-IIIT Pets classification task, which requires distinguishing subtle visual differences between classes, iiANET-L achieves the highest top-1 accuracy of 76.23%, outperforming some strong baselines including CrossViT-B (73.1%) and DeiT-B (71.0%). The smaller variant iiANET-B, achieves 74.04%, outperforming ViT-B/16 (51.23%) and ResNet-101 (59.25%), demonstrating strong generalization and fine-grained discriminative power.

**Analysis.** We observe that iiANET outperforms some CNN and transformer-based baselines across these datasets, achieving a better balance between accuracy and computational cost. Notably, ViT variants underperform on smaller datasets like AID and Oxford-IIIT Pets, likely due to limited training samples and the need for large-scale data to learn robust representations. In contrast, iiANET's hybrid backbone effectively captures both local details and long-range dependencies, providing improved generalization and fine-grained discriminative power.

### 4.3.2 Object Detection

We evaluate iiANET as a backbone for YOLOv8 on the COCO val2017 and test2017 datasets (Table 3). Both iiANET-(B, L) show good performance, achieving bounding box mAP of 62.6% and 63.1% on val2017, and 63.0% and 64.4% on test2017, respectively. Compared to backbones like ResNet-50 and Swin variants, iiANET achieves comparable or better accuracy with slightly lower computational cost, highlighting its efficiency in capturing multi-scale features and long-range dependencies. These results indicate that iiANET effectively balances precision, recall, and computational efficiency for object detection in complex scenes.

### 4.3.3 Instance Segmentation

We evaluate iiANET on COCO val2017 for instance segmentation using Mask R-CNN (1x schedule) (Table 4). Both iiANET-B and iiANET-L achieve better results compared to some CNN and transformer backbones, with $AP^{bb}$ of 45.3% and 45.8% and $AP^{m}$ of 39.5% and 42.1%. The results show iiANET ability to capture long-range dependencies and accurately segment instances in images with complex structures and occlusions.

### 4.3.4 Semantic Segmentation

Table 5 compares semantic segmentation (SS) performance on the ADE20K dataset across various backbones using the UPerNet framework. Traditional CNN-based models such as ResNet-101 with DeepLab v3+ or UPerNet achieve mIoUs below 45%, while transformer-based and hybrid architectures show clear gains. Swin-B and Focal-B improve SS to 49% mIoU but at the cost of over 120M parameters. In contrast, the proposed iiANET models achieve better accuracy with notably fewer parameters: iiANET-B attains 48.5% mIoU with 66M parameters, and iiANET-L achieves 49.2% mIoU with 92 M, outperforming some heavier transformer baselines. This demonstrates iiANET's strong balance between efficiency and SS performance.

Table 2: Classification result comparison on ImageNet-1K, AID, and Oxford-III datasets.

| | Backbone | Size | Train | Params | FLOPs | Top-1 | Top-5 |
|---|---|---|---|---|---|---|---|
| **IMAGENET-1K Deng et al. (2009)** | ResNet-101 He et al. (2016) | $224^2$ | 90 | 44.5M | 14.58G | 78.0% | 94.0% |
| | EffNet-B5 Tan & Le (2019) | $224^2$ | 90 | 30.4M | 4.49G | 83.6% | 96.7% |
| | Dense201 Huang et al. (2017) | $224^2$ | 90 | 20.0M | 7.35G | 77.42% | 93.6% |
| | ViT-L/16 Dosovitskiy et al. (2020) | $224^2$ | 150 | 304.3M | 59.69G | 76.53% | 93.2% |
| | MobileViT-S Mehta & Rastegari (2021) | $256^2$ | 90 | 6M | 2G | 77.0% | 94.6% |
| | DilateFormer-B Jiao et al. (2023) | $224^2$ | 120 | 48M | 9.96G | 84.9% | - |
| | BoT50 Srinivas et al. (2021) | $256^2$ | 90 | 25.6M | 3.18G | 84.4% | - |
| | CoAtNet-3 Dai et al. (2021b) | $224^2$ | 90 | 168M | 32.53G | 84.5% | - |
| | Vim-S Zhu et al. (2024) | $224^2$ | 90 | 26M | 5.3G | 80.3% | - |
| | S4ND-ViT-B Nguyen et al. (2022) | $224^2$ | 90 | 89M | 17.1G | 80.4% | - |
| | VMamba-T Liu et al. (2025) | $224^2$ | 90 | 30M | 4.9G | 82.6% | - |
| | Swin-T Dai et al. (2021b) | $224^2$ | 300 | 29M | 4.5G | 81.3% | - |
| | DeiT-B Touvron et al. (2021) | $224^2$ | 120 | 86M | 17.5G | 81.8% | - |
| | Cross-ViT-B Chen et al. (2021b) | $224^2$ | 120 | 105M | 20.1G | 82.2% | - |
| | CvT-21 Liu et al. (2021a) | $224^2$ | 300 | 32M | 7.1G | 82.5% | - |
| | Next-ViT-B Li et al. (2022) | $224^2$ | 300 | 44.8M | 8.3G | 83.2% | - |
| | iiANET-B | $299^2$ | 90 | 25.2M | 8.22G | 79.34% | 94.71% |
| | iiANET-L | $299^2$ | 120 | 50.9M | 13.45G | 84.9% | 96.83% |
| **AID Xia et al. (2017)** | ResNet-101 He et al. (2016) | $224^2$ | 90 | 44.5M | 14.58G | 68.93% | 93.37% |
| | EffNet-B5 Tan & Le (2019) | $224^2$ | 90 | 30.4M | 4.49G | 65.73% | 92.07% |
| | Dense201 Huang et al. (2017) | $224^2$ | 90 | 20.0M | 7.35G | 71.60% | 94.37% |
| | ViT-B/16 Dosovitskiy et al. (2020) | $224^2$ | 150 | 86.6M | 16.86G | 69.93% | 93.27% |
| | MobileViT-S Mehta & Rastegari (2021) | $256^2$ | 90 | 6M | 2G | 66.76% | 91.23% |
| | DiNAT-B Hassani & Shi (2022) | $224^2$ | 90 | 90M | 13.7G | 79.12% | 93.27% |
| | BoT50 Srinivas et al. (2021) | $256^2$ | 90 | 25.6M | 3.18G | 72.50% | 94.27% |
| | CoAtNet-3 Dai et al. (2021b) | $224^2$ | 90 | 168M | 32.53G | 80.17% | 94.93% |
| | Vim-S Zhu et al. (2024) | $224^2$ | 90 | 26M | 5.3G | 74.2% | - |
| | VMamba-T Liu et al. (2025) | $224^2$ | 90 | 30M | 4.9G | 78.9% | - |
| | Swin-T Dai et al. (2021b) | $224^2$ | 300 | 29M | 4.5G | 78.0% | - |
| | DeiT-B Touvron et al. (2021) | $224^2$ | 120 | 86M | 17.5G | 81.2% | - |
| | Cross-ViT-B Chen et al. (2021b) | $224^2$ | 120 | 105M | 20.1G | 80.6% | - |
| | iiANET-B | $299^2$ | 90 | 25.2M | 8.22G | 80.57% | 95.67% |
| | iiANET-L | $299^2$ | 90 | 50.9M | 13.45G | 83.11% | 96.07% |
| **OXFORD-III Parkhi et al. (2012)** | ResNet-50 He et al. (2016) | $224^2$ | 90 | 25.6M | 7.71G | 59.07% | 86.61% |
| | ResNet-101 He et al. (2016) | $224^2$ | 90 | 44.5M | 14.58G | 59.25% | 87.38% |
| | EffNet-B5 Tan & Le (2019) | $224^2$ | 90 | 30.4M | 4.49G | 47.54% | 78.92% |
| | Dense201 Huang et al. (2017) | $224^2$ | 90 | 20.0M | 7.35G | 62.55% | 86.84% |
| | ViT-B/16 Dosovitskiy et al. (2020) | $224^2$ | 150 | 86.6M | 16.86G | 51.23% | 82.31% |
| | ViT-L/16 Dosovitskiy et al. (2020) | $224^2$ | 150 | 304.3M | 59.69G | 53.19% | 83.28% |
| | MobileViT-S Mehta & Rastegari (2021) | $256^2$ | 90 | 6M | 2G | 51.89% | 84.52% |
| | DiNAT-B Hassani & Shi (2022) | $224^2$ | 90 | 90M | 13.7G | 69.37% | 92.09% |
| | DilateFormer-B Jiao et al. (2023) | $224^2$ | 120 | 48M | 9.96G | 64.85% | 88.18% |
| | BoT50 Srinivas et al. (2021) | $256^2$ | 90 | 25.6M | 3.18G | 64.13% | 90.00% |
| | CoAtNet-3 Dai et al. (2021b) | $224^2$ | 90 | 168M | 32.53G | 67.57% | 91.36% |
| | VMamba-T Liu et al. (2025) | $224^2$ | 90 | 30M | 4.9G | 67.8% | - |
| | Swin-T Dai et al. (2021b) | $224^2$ | 300 | 29M | 4.5G | 72.3% | - |
| | DeiT-B Touvron et al. (2021) | $224^2$ | 120 | 86M | 17.5G | 71.0% | - |
| | Cross-ViT-B Chen et al. (2021b) | $224^2$ | 120 | 105M | 20.1G | 73.1% | - |
| | iiANET-B | $299^2$ | 90 | 25.2M | 8.22G | 74.04% | 93.98% |
| | iiANET-L | $299^2$ | 90 | 50.9M | 13.45G | 76.23% | 94.54% |

Table 3: Object detection results on the COCO dataset.

| Backbone | Object Detector | $AP^b$ val2017 | $AP^b$ test2017 |
|---|---|---|---|
| ResNet-50 Carion et al. (2020) | Faster R-CNN | 36.7 | 37.9 |
| FD-SwinV2-G Wei et al. (2022) | HTC++ | - | 64.2 |
| Florence-CoSwin-H Yuan et al. (2021) | DyHead | 62.0 | 62.4 |
| Swin-L Liu et al. (2021b) | DINO | 63.2 | 63.3 |
| BEiT-3 Wang et al. (2022) | ViTDet | - | 63.7 |
| Swin V2-G Liu et al. (2022) | HTC++ | 62.5 | 63.1 |
| iiANET-B | YOLOv8 | 62.6 | 63.0 |
| iiANET-L | YOLOv8 | 63.1 | 64.4 |

Table 4: Instance Segmentation on COCO dataset with Mask R-CNN (1x schedule).

| Backbone | $AP^{box}$ | $AP^{box}_{50}$ | $AP^{box}_{75}$ | $AP^{mask}$ | $AP^{mask}_{50}$ | $AP^{mask}_{75}$ |
|---|---|---|---|---|---|---|
| ResNet-50 He et al. (2016) | 38.0 | 58.6 | 41.4 | 34.4 | 55.1 | 36.7 |
| PVT-M Hassani et al. (2023) | 42.0 | 64.4 | 45.6 | 39.0 | 61.6 | 42.1 |
| TRT-ViT-C Wang et al. (2021a) | 44.7 | 66.9 | 48.8 | 40.8 | 63.9 | 44.0 |
| Focal-T Xia et al. (2022) | 44.8 | 67.7 | 49.2 | 41.0 | 64.7 | 44.2 |
| UniFormer-S/h14 Yang et al. (2021a) | 45.6 | 68.1 | 49.7 | 41.6 | 64.8 | 45.0 |
| Swin-T Li et al. (2023) | 42.2 | 64.6 | 46.2 | 39.1 | 61.6 | 42.0 |
| Dilate-S Jiao et al. (2023) | 45.8 | 68.2 | 50.1 | 41.7 | 65.3 | 44.7 |
| BoT50 Srinivas et al. (2021) | 43.7 | - | - | 37.9 | - | - |
| PVT-L Wang et al. (2021b) | 42.9 | 65.0 | 46.6 | 39.5 | 61.9 | 42.5 |
| CAE-GReaT Zhang et al. (2024) | - | - | - | 44.0 | 67.9 | 47.3 |
| iiANET-B | 45.3 | 65.1 | 49.8 | 39.5 | 58.9 | 58.0 |
| iiANET-L | 45.8 | 68.3 | 51.7 | 42.1 | 65.1 | 59.5 |

## 4.4 Visualizing long-range spatial transport

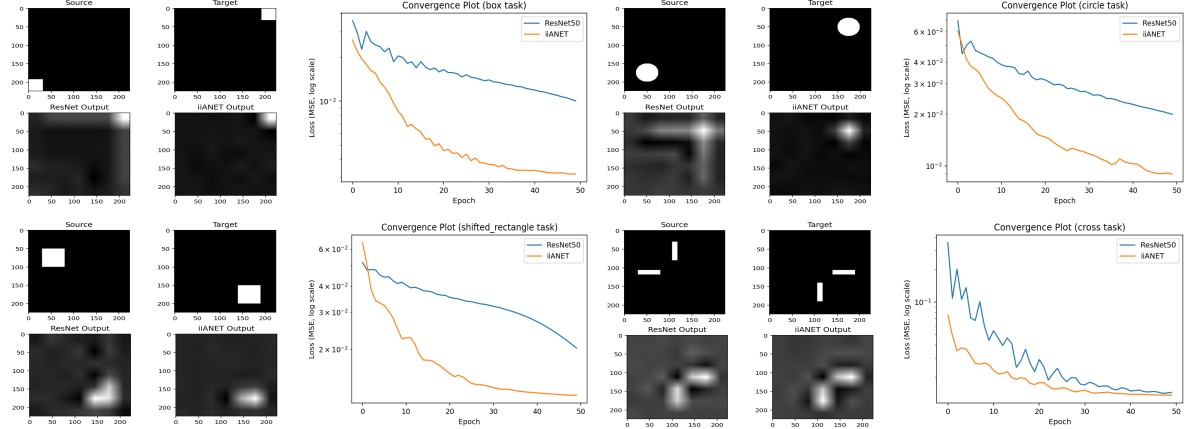

Figure 6: To evaluate the models' ability to capture long-range dependencies, we transport information across distant spatial regions. We place the source in one corner and the target in another corner using various shapes. Both ResNet-50 and iiANET are trained under identical settings (AdamW, lr=0.001, 50 epochs, input $224^2$, MSE loss). iiANET achieves faster, smoother convergence and more accurate reconstruction than ResNet-50, effectively capturing long-range dependencies and spatial transport patterns.

Table 5: Semantic segmentation results on the ADE20K dataset.

| Backbone | Method | img size | #param | mIoU % |
|----------|--------|----------|--------|--------|
| ResNet-101 He et al. (2016) | DeepLab v3+ | $512^2$ | 63M | 44.1 |
| ResNet-101 He et al. (2016) | UperNet | $512^2$ | 86M | 44.9 |
| DeiT-S Touvron et al. (2021) | UperNet | $512^2$ | 43M | 44.0 |
| Swin-B Dai et al. (2021b) | UperNet | $512^2$ | 121M | 48.1 |
| ViM-S Zhu et al. (2024) | UperNet | $512^2$ | 46M | 44.9 |
| Focal-B Yang et al. (2021b) | UperNet | $512^2$ | 126 | 49.0 |
| MambaVision-B Hatamizadeh & Kautz (2024) | UperNet | $512^2$ | 126 | 49.1 |
| iiANET-B | UperNet | $512^2$ | 66M | 48.5 |
| iiANET-L | UperNet | $512^2$ | 92M | 49.2 |

## 4.5 Ablation Studies

We performed ablation studies to analyze the impact of different components, scaling, and efficiency of iiANET.

**Settings.** All ablation experiments were conducted on the AID dataset Xia et al. (2017) using an NVIDIA GeForce RTX 2070 with Max-Q Design (8 GB VRAM), CUDA 12.1 and PyTorch 2.0.

### 4.5.1 iiABlock vs. MHSA Complexity Analysis

Table 6 shows *iiABlock* and *MHSA* evaluation across channel-spatial scaling, and attention heads.

- **Channel Scaling (32→64→128 @ 112×112):** iiABlock shows more efficient scaling with channel width, achieving lower FLOPs ($-5.4\%$), fewer parameters ($-25.4\%$), and $\approx 28\%$ faster inference compared to MHSA. Its memory growth is also smaller, confirming better computational efficiency.

- **Spatial Resolution (28×28→14×14 @ 512/1024):** iiABlock achieves $\approx 30\%$ lower FLOPs, 29% fewer parameters, and 20% faster latency than MHSA. It also uses slightly less memory and maintains higher throughput, demonstrating robust efficiency across scales.

- **Heads (2→4 @ 56×56):** With 2× heads, iiABlock maintains about 4% lower FLOPs and 21% fewer parameters, while achieving up to $+34\%$ higher throughput than MHSA. Although latency increases slightly, iiABlock consistently achieves faster inference and better scaling efficiency.

Table 6: Complexity comparison between iiABlock and MHSA across scaling dimensions.

| Type | Spatial Dim. | In/Out | FLOPs(G) | Params | Latency (ms) | TP(img/s) | Head |
|------|-------------|--------|----------|--------|--------------|-----------|------|
| **Scaling Channels** | | | | | | | |
| iiABlock | (112,112) | 32/64 | 1.024 | 5.932K | 38.87 | 25.73 | 1 |
| iiABlock | (112,112) | 64/128 | 1.230 | 21.944K | 41.32 | 24.20 | 1 |
| MHSA | (112,112) | 32/64 | 1.040 | 7.520K | 53.93 | 18.54 | 1 |
| MHSA | (112,112) | 64/128 | 1.317 | 29.376K | 57.90 | 17.27 | 1 |
| **Scaling Attention Heads** | | | | | | | |
| iiABlock | (56,56) | 32/64 | 108.471M | 5.932K | 9.44 | 105.94 | 2 |
| iiABlock | (56,56) | 32/64 | 167.472M | 5.932K | 12.00 | 83.34 | 4 |
| MHSA | (56,56) | 32/64 | 112.586M | 7.520K | 12.64 | 79.11 | 2 |
| MHSA | (56,56) | 32/64 | 171.586M | 7.520K | 12.81 | 78.05 | 4 |
| **Scaling Spatial Dimension** | | | | | | | |
| iiABlock | (28,28) | 512/1024 | 1.037 | 1.305M | 5.59 | 178.74 | 4 |
| iiABlock | (14,14) | 512/1024 | 257.627M | 1.305M | 4.14 | 241.35 | 4 |
| MHSA | (28,28) | 512/1024 | 1.454 | 1.841M | 6.61 | 151.29 | 4 |
| MHSA | (14,14) | 512/1024 | 361.842M | 1.841M | 4.32 | 231.71 | 4 |

### 4.5.2 Effect of iiABlock Components

Table 7 highlights the effectiveness of the proposed combination of modules within the iiABlock. Variant (c), which integrates MBConv, Dilated Convolution, and MHSA, achieves the best performance with a top-1 accuracy of 80.57%. This demonstrates that the synergy between local depth-wise convolutions, dilated receptive fields, and global attention significantly improves feature representation. Variants (a), (b), and (e) show progressive improvements, while (d), which lacks MBConv, suffers a performance drop, underscoring the importance of lightweight local feature extraction.

Table 7: Ablation studies on various variants of iiABlock using the AID dataset.

| Settings | Model (Components) | Size | Top-1 |
|----------|-------------------|------|-------|
| (a) | MBConv2 | 299 | 69.23% |
| (b) | MBConv + Dilated Conv | 299 | 74.85% |
| (c) | MBConv + Dilated Conv + MHSA | 299 | **80.57%** |
| (d) | Dilated + MHSA | 299 | 67.01% |
| (e) | MBConv + MHSA | 299 | 78.72% |

### 4.5.3 Ablation on MHSA Heads and Block Ratio

Table 8 presents the results of ablation experiments in which the channel ratio was modified from 1.6.1 to 2.4.2, resulting in a reduction of 9.6M parameters and 2.97G FLOPs, with a corresponding 6.36% decrease in top-1 accuracy. Increasing the number of MHSA heads in the last iiANET block from 8 to 16 caused minimal changes in computational cost and a -0.34% change in accuracy. These results indicate that reducing the block width improves efficiency but decreases accuracy, while increasing attention heads has minimal effect, suggesting the original configuration is near optimal.

Table 8: Ablation study on the effect of changing the MHSA head size and iiABlock ratio on AID dataset.

| Settings | Model (Components) | Size | #params | FLOPs | Top-1 |
|---|---|---|---|---|---|
| iiANET *(all layers)* | Ratio: 1.6.1 → 2.4.2 | 299 | 15.72M | 5.64G | 74.21% |
| iiANET *(layer4)* | Head Size: 8 → 16 | 299 | 25.32 | 8.604G | 80.23% |

### 4.5.4 Effects of Different Fusion Strategies

Table 9 compares cross attention, gated fusion, and additive fusion in iiANET-B. Additive fusion provides a better trade-off, with lower computational cost (8.60 GFLOPs), fewer parameters 25.32M, and faster inference (27.70 ms, 36.10 img/s), with a slight decrease in top-1 accuracy 80.57% compared to gated fusion. Cross attention achieves higher accuracy 81.32% but at the cost of increased complexity and latency.

Table 9: Ablation studies on fusion

| Fusion Type | #params | FLOPs | latency(m/s) | TP(img/sec) | Top-1 (%) |
|---|---|---|---|---|---|
| Cross Attention Chen et al. (2021c) | 27.18M | 9.39G | 42.95 | 23.29 | 81.32% |
| Gated fusion Jiang & Ji (2022) | 28.422M | 9.56G | 36.66 | 27.28 | 80.75 % |
| Additive Fusion | 25.32M | 8.60G | 27.70 | 36.10 | 80.57% |

### 4.5.5 Effect of register tokens

Table 10 demonstrates that introducing register tokens into the 2D-MHSA improves performance on the AID dataset. With four registers, the model achieves the highest top-1 accuracy of 80.57%, compared to 80.36% without any. The consistent improvement suggests that registers help compensate for MHSA's order-agnostic nature by enhancing the contextual richness of learned representations.

Table 10: Ablation study on the effect of adding register tokens to 2D-MHSA mechanism

| Registers | Top-1 |
|---|---|
| 0 | 80.36% |
| 1 | 80.37% |
| 2 | 80.45% |
| 4 | **80.57%** |

Figure 7: Grad-CAM Selvaraju et al. (2017) heatmaps illustrating the effect of registers on iiANET for the AID dataset Xia et al. (2017). Adding registers to the r-MHSA module enables iiANET to capture long-range dependencies more effectively, improving interpretability of the learned representations.

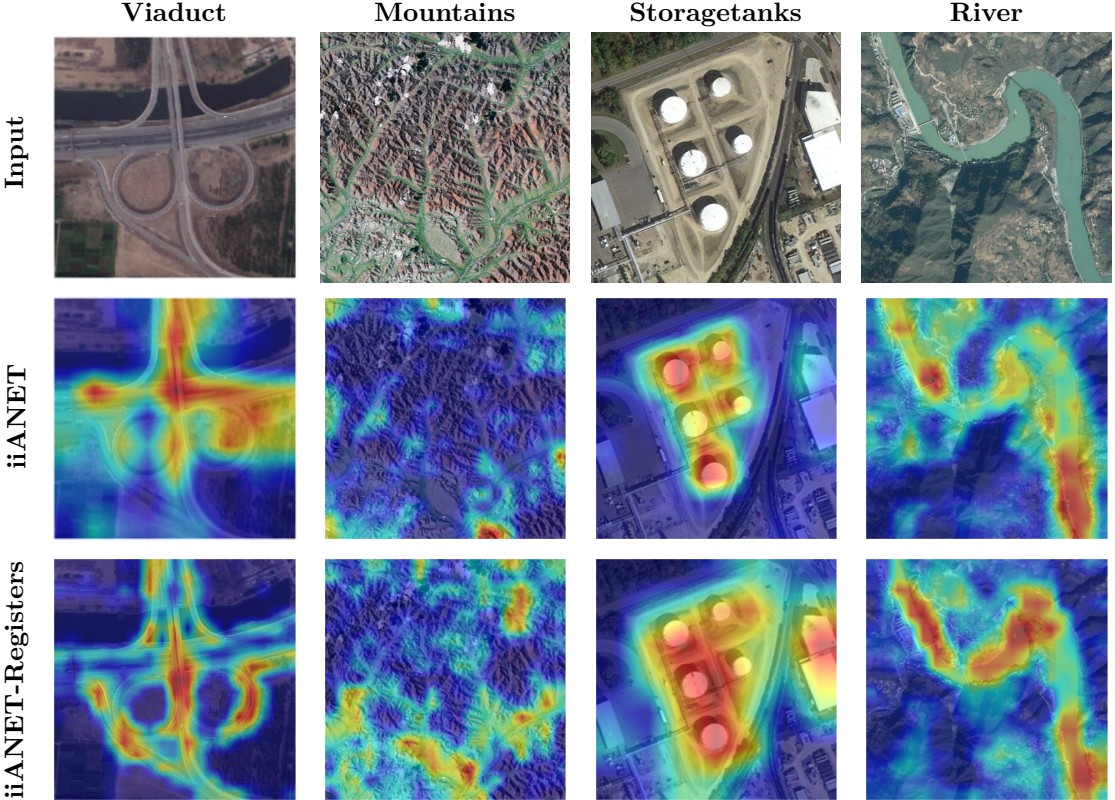

## 5 Conclusion

This work proposed iiANET, a novel hybrid model designed to efficiently improve long-range dependencies in complex images by integrating CNN layers and the MHSA mechanism with registers in parallel. Comprehensive qualitative and quantitative results show improvements in capturing long-range dependencies compared to some previous SOTA models. Additionally, we validate the performance of our model across diverse datasets and highlight its potential as an efficient backbone for visual downstream tasks.

### 5.1 Limitations

iiANET is effective on images with prevalent long-range dependencies but may be less effective on datasets with mostly localized objects. Its multi-branch and ratio design can also introduce scaling challenges, making it difficult to determine optimal configurations in some downstream visual recognition tasks.

**Acknowledgment** This work was supported by NewraLab, Suzhou, China, an AI research and development startup founded by Yunusa Haruna. The authors gratefully acknowledge the support of the NewraLab team throughout this research.

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
