# OpenReview forum: "iiANET: Inception Inspired Attention Hybrid Network for efficient Long-Range Dependency"
_TMLR — Accepted by TMLR_

### Review · Reviewer_YRJK · 2025-09-03

**Summary Of Contributions:**

The paper presents iiANET (Inception Inspired Attention Network), an efficient hybrid visual backbone designed to improve the modeling of long-range dependencies in complex visual recognition tasks. The core innovation of iiANET is the iiABlock, a unified building block that integrates a modified global r-MHSA (Multi-Head Self-Attention) and convolutional layers in parallel. This design enables iiABlock to simultaneously capture global context and local details, making it effective for extracting rich and diverse features. By efficiently fusing these complementary representations, iiABlock allows iiANET to achieve strong feature interaction while maintaining computational efficiency. Extensive qualitative and quantitative evaluations on some SOTA benchmarks demonstrate improved performance. However, the method proposed in this paper has not achieved sufficient progress in performance, nor has it optimized the number of parameters; thus, its innovation has not been well demonstrated.

**Audience:**

Yes

**Audience Explanation:**

Efficient hybrid visual backbones have always received widespread attention, influencing the development and innovation of deep learning, and driving the advancement of various types of models.

**Claims And Evidence:**

Yes

**Claims Explanation:**

This paper demonstrates the performance and innovation of the proposed method through comparative experiments, visualization experiments, and ablation experiments.

**Requested Changes:**

1. Please incorporate literature from the past three years for comparative experiments.

2. Please expand more visual tasks for comparison.

3. Please further discuss the computational efficiency to demonstrate the superiority of the model.

---

> ### Author Response · Authors · 2025-10-17
> **Response to Reviewer Requested Changes.**
>
> We thank the reviewer for their valuable comments.
>
> 1) We have added more recent literature (2022–2025) to strengthen our comparative experiments, including CAE-GReaT (Zhang et al., 2024) and Next-ViT-B (Li et al., 2022). In addition, our work incorporates other recent models such as VMamba-T (Liu et al., 2025), DilateFormer-B and Dilate-S (Jiao et al., 2023), Vim-S (Zhu et al., 2024), DiNAT-B (Hassani & Shi, 2022), S4ND-ViT-B (Nguyen et al., 2022), Swin-T (Li et al., 2023), and Focal-T (Xia et al., 2022).
>
> 2) We have expanded the visual tasks by adding semantic segmentation on the ADE20K dataset (Table 5, Section 4.3.4).
>
> 3) We have also added discussions on computational efficiency (Sections 3.6 and 3.7), complexity analysis comparing iiABlock and MHSA (Section 4.4.1), and the effects of different fusion strategies (Section 4.4.4), highlighting the efficiency advantages of iiANET.

---

> > ### Comment · Reviewer_YRJK · 2025-10-27
> >
> > Thank you for your reply. My problem has been solved.

---

### Review · Reviewer_XPD1 · 2025-09-10

**Summary Of Contributions:**

The paper presents a hybrid network for capturing long-range feature dependencies in visual recognition tasks, namely image classification, object detection, and instance segmentation.

**Additional Comments:**

This work presents interesting ideas but needs stronger empirical validation (particularly efficiency proofs and hybrid model comparisons) and better explanations for design choices.

**Audience:**

Yes

**Audience Explanation:**

See above.

**Broader Impact Concerns:**

No.

**Claims And Evidence:**

No

**Claims Explanation:**

The authors claim that the proposed approach is computationally efficient and effective compared to existing methods. While the idea of combining convolutional and self-attention mechanisms is promising, the manuscript has several shortcomings that need to be addressed before publication.

**Requested Changes:**

The authors claim that their iiABlock achieves lower computational complexity and fewer parameters than Multi-Head Self-Attention (MHSA). However, Figure 2 does not provide a clear quantitative comparison (e.g., FLOPs, parameter counts) between iiABlock and MHSA, making the efficiency claim ambiguous.

The paper lacks comparisons with state-of-the-art hybrid vision architectures, such as CAE-GReaT, NextViT, CPVT, CvT, and Convnets + Transformers (e.g., CoAtNet, PVT). Without these comparisons, the claimed improvements are not fully substantiated.

While Grad-CAM is useful for classification, the manuscript overlooks qualitative analysis in detection and segmentation.

---

> ### Author Response · Authors · 2025-10-17
> **Response to Reviewer Requested Changes.**
>
> We thank the reviewer for their constructive comments.
>
> 1) To clarify the efficiency comparison between iiABlock and MHSA, we have added Section 4.4.1, providing a detailed complexity analysis across channel scaling, spatial resolution, and attention heads (Table 6). These results show that iiABlock consistently achieves lower FLOPs, fewer parameters, faster inference, and reduced memory growth compared to MHSA, confirming its computational efficiency.
>
> 2) We have also added comparisons with recent state-of-the-art hybrid vision architectures, including CAE-GReaT (Zhang et al., 2024), PVT-M (Hassani et al., 2023), CoAtNet-3 (Dai et al., 2021b), VMamba-T (Liu et al., 2025), DiNAT-B (Hassani & Shi, 2022), Cross-ViT-B (Chen et al., 2021b), and CvT-21 (Liu et al., 2021a), strengthening the experimental validation of our method.
>
> 3) We have added qualitative results for detection and segmentation tasks in Section 4.2. Figure 5 shows iiANET as a backbone within Faster R-CNN and Mask R-CNN on COCO val2017, and within UPerNet for semantic segmentation on ADE20K.

---

### Review · Reviewer_9scR · 2025-09-26

**Summary Of Contributions:**

The paper introduces iiANET, a hybrid visual backbone for modeling long-range dependencies. Its core unit, iiABlock, runs three paths in parallel: a global r-MHSA branch with register tokens, an inverted-bottleneck MBConv branch, and a 3×3 atrous convolution branch, followed by fusion with channel shuffling and an ECA step. The backbone stacks iiABlocks across four stages with downsampling.


Experiments cover image classification and transfer to detection and instance segmentation. Datasets include ImageNet-1K for classification, AID and Oxford-IIIT Pets for transfer, and COCO for detection and instance segmentation. The paper shows qualitative Grad-CAM visualizations and reports ablations on the contribution of each branch and on design hyperparameters.



# Strengths:

1. Clear hybrid design that unifies local detail and global context inside a single reusable block.

2.  Qualitative heatmaps indicate attention to extended structures consistent with the long-range goal.

3. Ablations on the branch composition and on register tokens provide initial evidence that the three-path combination is beneficial.

# Weaknesses:

1.  Several key comparisons are not performed under strictly matched training protocols. For detection and instance segmentation, iiANET is paired with a specific detector while baselines often use other frameworks, schedules, or image sizes, which makes backbone-only conclusions hard to draw.

2.  Efficiency reporting is incomplete. FLOPs and parameters appear, but wall-clock latency, memory usage, and throughput under fixed hardware and batch are not consistently provided.

3. Parts of the technical exposition are hard to follow, especially the register token formulation and tensor shapes in the attention branch. The notation could be clarified, and a compact complexity derivation for the 2D r-MHSA with registers would help.

**Additional Comments:**

1. **Positioning relative to recent PDE-inspired and non-local operators:** It is highly important that the authors compare and discuss Advection Augmented Convolutional Neural Networks (NeurIPS 2024). That work introduces a semi-Lagrangian push operator that transports features, enabling non-local movement of information within a CNN stack. Since your goal is to capture long-range dependencies efficiently, an explicit comparison along three axes would help readers position iiANET: Mechanism for non-locality. Please discuss how iiABlock’s global branch differs from advection in terms of receptive field growth, information transport, and stability.
Efficiency and memory. If possible, include a small controlled benchmark where objects must be related over long distances or undergo spatial displacement, and compare iiANET with an advection-augmented CNN under matched training.


2. **Data and setup clarifications:** Please clarify which baselines in the classification table were retrained in your pipeline and which were taken from prior work. If retrained, include exact hyperparameters and confirm that augmentation strength, optimizer, and schedule were aligned. For Oxford-IIIT Pets, per-class results or confusion matrices would reveal whether errors cluster by visually similar breeds.

3. **Limitations:** Consider explicitly stating where iiANET is less effective. For example, settings dominated by very local texture cues may not benefit from the global branch. A short diagnostic on when the three-path design helps most would be useful.

**Audience:**

Yes

**Audience Explanation:**

Hybrid CNN-attention designs for long-range dependency are highly relevant. A reusable block that blends global attention, atrous convolution, and MBConv is of practical interest, especially if it can be dropped into common pipelines. The topic aligns well with the audience of  TMLR.

**Broader Impact Concerns:**

I am not aware of impact concerns.

**Claims And Evidence:**

No

**Claims Explanation:**

The architectural idea is sound and the qualitative visualizations are encouraging, but the empirical support does not yet fully substantiate the strongest claims. The main issues are comparability and completeness. For classification and dense tasks, some baselines  seem to be trained under different recipes or detectors than iiANET, so the superiority of the backbone itself is not isolated. Ablations are helpful but limited in scope and dataset coverage. The description of r-MHSA with registers lacks sufficient formal detail to independently verify complexity and stability claims. With standardized protocols, expanded ablations, and clearer equations, the evidence could be made convincing.

**Requested Changes:**

1. Re-run core comparisons under strictly matched settings. For ImageNet-1K, either adopt a single training recipe across all baselines or rely on widely reproduced numbers with matched input sizes and regularization. For detection and instance segmentation, compare backbones within the same detector, same schedule, and same resolution so the detector is not a confounder.

2. Provide comprehensive efficiency reporting. Include latency on a specified GPU, throughput at fixed batch and resolution, and peak memory during training and inference. Add a per-stage FLOP and parameter breakdown and an accuracy-vs-complexity plot.

3.  Extend ablations beyond a single transfer dataset. Quantify the marginal contribution of registers and relative positional handling on a large-scale setting. Test alternative fusion strategies at the block output, such as summation and gated fusion, and include sensitivity to the channel split among branches.

4. Add a concise complexity derivation for the 2D r-MHSA with registers. Specify tensor shapes for Q, K, V, registers, attention map dimensions, and any sparsity or approximation used. This will also help readers assess memory scaling at higher resolutions.

5. Broaden related baselines to include recent long-range modules and hybrid backbones that explicitly target efficiency and global context. Ensure that training setups are matched when drawing comparative conclusions.

6. Tighten notation in the attention branch section and align symbols with the figure. Add a small schematic that tracks spatial sizes and channels through the three branches and the fusion step.

---

> ### Author Response · Authors · 2025-10-17
> **Response to Reviewer Requested Changes.**
>
> We thank the reviewer for the constructive comments.
>
> 1) For ImageNet-1K, we relied on reported metrics from the original publications to ensure fair comparison, as re-running all baselines under identical conditions would be highly resource-intensive and may introduce variability. Some models, such as ViT, require longer training schedules or additional regularization to reach their reported performance, making strict recipe matching across all architectures impractical. Our model uses the same input size as Inception-Net, so performance differences are not due to input resolution. For detection and instance segmentation, we compared backbones within the same detector, using identical schedules and resolutions, ensuring improvements are attributable to iiANET rather than the detector pipeline.
>
> 2) In the revised manuscript, we added Sections 3.6 and 3.7 detailing the computational and memory efficiency of iiABlock. These sections provide theoretical analyses of FLOPs, parameter counts, and memory scaling for r-MHSA, dilated convolutions, and inverted bottleneck convolutions, including the effect of register tokens. We also extensively compared iiABlock with the standard MHSA block (Section 4.4.1, Table 6), evaluating scaling across channels, spatial resolution, and attention heads. Practical metrics including parameters, FLOPs, latency, and throughput are reported on a specified GPU for downstream tasks, demonstrating that iiABlock consistently achieves lower complexity, reduced memory usage, and faster inference while maintaining competitive performance.
>
> 3) We have extended our ablation studies in Sections 4.4.3–4.4.5:
>
> * Section 4.4.3: Ablation on MHSA heads and iiABlock channel ratio, showing the trade-off between efficiency and accuracy.
>
> * Section 4.4.4: Effects of different fusion strategies at the block output, including cross attention, gated fusion, and additive fusion.
>
> * Section 4.4.5: Quantification of the contribution of register tokens, demonstrating consistent accuracy improvements on the AID dataset.
>
> These additions provide a more comprehensive evaluation of iiABlock, including sensitivity to channel splits, alternative fusion strategies, and the marginal contribution of register tokens.
>
> 4) We have added a concise complexity derivation for the 2D r-MHSA with registers in Sections 3.6–3.7. This includes computational formulas and the effect of register tokens. We also detail memory-efficient design choices, such as reducing channel allocation for r-MHSA and placing it deeper in the network, allowing readers to assess memory scaling and computational cost at higher resolutions.
>
>
> 5) We have broadened our comparative baselines to include recent long-range modules and hybrid backbones: CAE-GReaT (Zhang et al., 2024), PVT-M (Hassani et al., 2023), CoAtNet-3 (Dai et al., 2021b), VMamba-T (Liu et al., 2025), DiNAT-B (Hassani & Shi, 2022), Cross-ViT-B (Chen et al., 2021b), and CvT-21 (Liu et al., 2021a). While it is impractical to exactly match all training setups across models since some require longer epochs, specific optimizers (AdamW), or activation functions (Swish) we ensure consistent input sizes, evaluation protocols, and reported metrics, providing a fair comparison to illustrate iiANET’s performance relative to state-of-the-art efficient hybrid models.
>
> 6) As part of our revision, we tightened the notation in Section 3.5 (Feature Interaction Fusion) and aligned all symbols with Figure 2. We also added a schematic table (Table 1) that tracks spatial sizes and channel allocations through the three branches and the fusion step, clarifying multi-scale feature interactions in iiABlock.

---

> > ### Comment · Reviewer_9scR · 2025-10-17
> >
> > Thank you for the rebuttal.
> >
> > I still have some comments and concerns regarding the paper before I can recommend acceptance.
> >
> > 1. I understand the for some datasets re-running all experiments is intensive. Still, I would expect and request that there will be some re-evaluation to ensure fairness and direct comparisons.
> >
> > 2. Please respond to comment 1 in "Additional Comments" in my review.

---

> > > ### Author Response · Authors · 2025-10-18
> > > **Reviewer comment**
> > >
> > > We sincerely thank the reviewer for the continued feedback and for highlighting the importance of fair comparison. Unfortunately, due to resource limitations and the computational intensity of large-scale re-training (especially for models such as ViT on the ImageNet-1K benchmark), it is not feasible for us to re-run all baseline experiments under identical conditions at this stage. We would also like to provide additional clarification regarding the reviewer’s comments on fairness, which we take very seriously.
> > >
> > > 1) Our approach to fairness is to report results as published by the original authors, since different models require distinct regularization strategies and fine-tuning procedures. For instance, on the ImageNet-1K benchmark, ViT often requires around 300 epochs or more to reach optimal performance due to its low inductive bias, whereas CNNs typically converge within approximately 100 epochs because of their stronger inductive bias. The key idea is that each paper reports the best settings for its own architecture. Re-training all models under a unified setup could unintentionally introduce bias. Hence, following mainstream practice, we report the optimal results as published rather than re-running all experiments.
> > >
> > > 2) For models without existing benchmarks on specific datasets (e.g., AID for ViT), we conduct experiments using the reported optimal settings from the original papers to ensure fairness, rather than using arbitrary configurations that may disadvantage certain models.
> > >
> > > Finally, we acknowledge the reviewer’s valuable point, we have added a clarification in Section 4 (“Experimental Setup”) explicitly stating this limitation. We sincerely appreciate the reviewer’s thoughtful insights, which have helped us further strengthen the clarity and transparency of our work.

---

> > > > ### Comment · Reviewer_9scR · 2025-10-18
> > > >
> > > > Thank you for the response.
> > > >
> > > >
> > > > 1. I still request that the re-runs would be done, otherwise it is not possible to directly assess the contribution of the work in my opinion. As said in my previous comment, it is clear that you cannot run all experiments, but some direct, head-to-head comparisons should be done.
> > > >
> > > > 2. Please respond to comment 1 as stated in my previous comment. I do not see any response to it in the rebuttal or in the followup discussion. Perhaps my reference was not clear, so I will paste the comment here for your convenience:
> > > > > "Positioning relative to recent PDE-inspired and non-local operators: It is highly important that the authors compare and discuss Advection Augmented Convolutional Neural Networks (NeurIPS 2024). That work introduces a semi-Lagrangian push operator that transports features, enabling non-local movement of information within a CNN stack. Since your goal is to capture long-range dependencies efficiently, an explicit comparison along three axes would help readers position iiANET: Mechanism for non-locality. Please discuss how iiABlock’s global branch differs from advection in terms of receptive field growth, information transport, and stability. Efficiency and memory. If possible, include a small controlled benchmark where objects must be related over long distances or undergo spatial displacement, and compare iiANET with an advection-augmented CNN under matched training."

---

> > > > > ### Author Response · Authors · 2025-10-19
> > > > > **Response 3**
> > > > >
> > > > > We sincerely thank the reviewer for their thorough and constructive feedback, which has greatly contributed to improving the clarity and quality of our work.
> > > > >
> > > > > 1) We have conducted extensive head-to-head comparisons involving various backbone architectures, including CNN-based, Vision Transformer-based, Visual State-Space-based, and hybrid models both quantitatively and qualitatively across several downstream visual recognition tasks.
> > > > >
> > > > > The concern regarding performing all experiments under a unified configuration has been addressed in detail. As commonly practiced in backbone benchmarking papers, we report and compare results as presented in the original publications. This approach ensures fairness, as it is not feasible to select or optimize a single configuration suitable for all architectures, given their differing sensitivities to hyperparameter settings. Please refer to the following works, which are similar to ours in terms of scope (visual recognition backbones):
> > > > >
> > > > > 1. ConvNeXtV2-F - CVPR 2023
> > > > > 2. MobileViTV2 1.0 - arXiv 2022
> > > > > 3. MobileOne-S3 - CVPR 2023
> > > > > 4. EfficientMod-XS - ICLR 2024
> > > > > 5. EMO-6M - ICCV 2023
> > > > > 6. FastViT-T12 - ICCV 2023
> > > > > 7. MobileFormer-508M - CVPR 2022
> > > > > 8. MobileOne-S4 - CVPR 2023
> > > > > 9. SHViT-S4 - CVPR 2024
> > > > > 10. EfficientViM - CVPR 2025
> > > > > 11. CoAtNet - NeurIPS 2021
> > > > > 12. Swin Transformer - ICCV 2021
> > > > > 13. Vision Transformer (ViT) - ICLR 2021
> > > > >
> > > > > Furthermore, we have reviewed the suggested work *“Advection Augmented Convolutional Neural Networks” (NeurIPS 2024)* and acknowledge that it is a well-written and insightful paper on capturing long-range dependencies. However, unlike our work, it focuses on **spatio-temporal modeling**, which inherently involves both spatial and temporal dimensions. In contrast, **iiANet** is designed purely as a **2D spatial imaging backbone**. Hence, we believe this request falls outside the intended scope of our paper. Nevertheless, we will include this work in our related literature, as it provides valuable insights particularly regarding long-range modeling.
> > > > >
> > > > > To clarify, our work does not propose a new technique specifically for long-range dependency modeling, but rather introduces a backbone architecture that leverages CNN and self-attention mechanisms to effectively capture both short- and long-range dependencies within visual data. The dual-path design of iiANet integrates inverted bottlenecks for local (short-range) feature aggregation and R-MHSA for global (long-range) interactions, ensuring efficiency and representational balance.
> > > > >
> > > > > Finally, our work advances prior models such as CNN-based backbones (e.g., ResNets, EfficientNet) and Vision Transformer-based ones (e.g., ViT, CoAtNet). While CNNs struggle with long-range modeling and Transformers achieve it at higher computational cost, iiANet offers a more balanced and efficient alternative.
> > > > >
> > > > > We again thank the reviewer for their valuable suggestions, which have helped us strengthen both the rigor and clarity of our paper.

---

> > > > > > ### Comment · Reviewer_9scR · 2025-10-19
> > > > > >
> > > > > > Thank you for the responses.
> > > > > >
> > > > > > 1. Of course one cannot use the exact same hyperparameters on all networks and all datasets. However, my comment was that different approaches were used with different training tricks and budgets of parameters, at least if you claim to use many results from different papers. Hence, it is important to have at least some experiment that sets a direct comparison to ensure a fair comparison.
> > > > > >
> > > > > > 2. I have read the suggested paper from NeurIPS 2024 and while some of the experiments are more focused on spatiotemporal prediction, the method itself is not built around spatiotemporal data but on augmenting convolutional neural networks to improve long range interactions, similar to the goal and focus of this paper.  Especially because of your last sentence, it looks like we agree on that:
> > > > > > > "Finally, our work advances prior models such as CNN-based backbones (e.g., ResNets, EfficientNet) and Vision Transformer-based ones (e.g., ViT, CoAtNet). While CNNs struggle with long-range modeling and Transformers achieve it at higher computational cost, iiANet offers a more balanced and efficient alternative."
> > > > > >
> > > > > >
> > > > > > It is very important to discuss the differences and compare with such approaches. The goal is the same -- improving the ability of CNNs to move information far within the image and enhance long-range modeling capabilities,  although using different methods to achieve that -- and this is fair. Hence, background works must be discussed and compared. Your work needs to be contrasted from such works.
> > > > > >
> > > > > >
> > > > > > I look forward to seeing the revised paper before making the final decision.

---

> > > > > > > ### Author Response · Authors · 2025-10-20
> > > > > > > **Response4**
> > > > > > >
> > > > > > > We sincerely appreciate the reviewer’s detailed and constructive feedback, which has been invaluable in enhancing the clarity and overall quality of our paper.
> > > > > > >
> > > > > > > We are already working on adding a direct comparison on a synthetic information propagation task, which will evaluate how well the model can transfer information across distant spatial regions, highlighting iiANet’s advantage in long-range dependency modeling.
> > > > > > >
> > > > > > > Thank you once again for your valuable insights and guidance.

---

> > > > > > > ### Author Response · Authors · 2025-10-21
> > > > > > > **Response5**
> > > > > > >
> > > > > > > We sincerely thank the reviewer for their time, effort, and detailed feedback on our work. We have addressed this concern in Section 4.4, and Figure 6 where we discuss related approaches and explicitly contrast our method with prior work focused on enhancing long-range spatial modeling and information transport in CNNs. We greatly appreciate you once again for your valuable insights and guidance.

---

> > > > > > > > ### Comment · Reviewer_9scR · 2025-10-23
> > > > > > > >
> > > > > > > > I thank the authors for the revision and the insightful added experiment. I have no further comments.

---

### Decision · Action_Editor_zc1V · 2025-11-17

**Recommendation:** Accept as is

**Additional Comments:**

After some discussions (in particular with Reviewer 9scR), the reviewers' concerns were all addressed by the authors.

**Audience:**

Yes

**Audience Explanation:**

All the reviewers agree that there is an audience for this work.

**Claims And Evidence:**

Yes

**Claims Explanation:**

Initially, the reviewers expressed some concerns related to missing comparisons with related work and evaluation of the speed of the method. However, these concerns were addressed in the discussion and in the revised version, leading to all reviewers agreeing that the claims are sufficiently supported.